

# The coefficient of determination R-squared is more informative than SMAPE, MAE, MAPE, MSE and RMSE in regression analysis evaluation

Davide Chicco[1], Matthijs J. Warrens[2] and Giuseppe Jurman[3]

[1] Institute of Health Policy Management and Evaluation, University of Toronto, Toronto, Canada
[2] Groningen Institute for Educational Research, University of Groningen, Groningen, Netherlands
[3] Data Science for Health Unit, Fondazione Bruno Kessler, Trento, Italy

## ABSTRACT

Regression analysis makes up a large part of supervised machine learning, and consists of the prediction of a continuous independent target from a set of other predictor variables. The difference between binary classification and regression is in the target range: in binary classification, the target can have only two values (usually encoded as 0 and 1), while in regression the target can have multiple values. Even if regression analysis has been employed in a huge number of machine learning studies, no consensus has been reached on a single, unified, standard metric to assess the results of the regression itself. Many studies employ the mean square error (MSE) and its rooted variant (RMSE), or the mean absolute error (MAE) and its percentage variant (MAPE). Although useful, these rates share a common drawback: since their values can range between zero and +infinity, a single value of them does not say much about the performance of the regression with respect to the distribution of the ground truth elements. In this study, we focus on two rates that actually generate a high score only if the majority of the elements of a ground truth group has been correctly predicted: the coefficient of determination (also known as $R$-squared or $R^2$) and the symmetric mean absolute percentage error (SMAPE). After showing their mathematical properties, we report a comparison between $R^2$ and SMAPE in several use cases and in two real medical scenarios. Our results demonstrate that the coefficient of determination ($R$-squared) is more informative and truthful than SMAPE, and does not have the interpretability limitations of MSE, RMSE, MAE and MAPE. We therefore suggest the usage of $R$-squared as standard metric to evaluate regression analyses in any scientific domain.

## INTRODUCTION

The role played by regression analysis in data science cannot be overemphasised: predicting a continuous target is a pervasive task not only in practical terms, but also at a conceptual level. Regression is deeply investigated even nowadays, to the point of still being worth of considerations in top journals (*Jaqaman & Danuser, 2006*; *Altman & Krzywinski, 2015*; *Krzywinski & Altman, 2015*), and widespread used also in the current

Corresponding author
Davide Chicco,
davide.chicco@gmail.com

scientific war against COVID-19 (*Chan et al., 2021*; *Raji & Lakshmi, 2020*; *Senapati et al., 2020*; *Gambhir et al., 2020*). The theoretical basis of regression encompasses several aspects revealing hidden connections in the data and alternative perspectives even up to broadly speculative view: for instance, interpreting the whole statistical learning as a particular kind of regression (*Berk, 2020*), or framing deep neural networks as recursive generalised regressors (*Wüthrich, 2020*), or even provocatively pushing such considerations to the limit of setting the whole of statistics under the regression framework (*Hannay, 2020*). The relevancy of the topic is clearly reflected in the wide and heterogeneous literature covering the different aspects and insights of the regression analysis, from general overviews (*Golberg & Cho, 2004*; *Freund, Wilson & Sa, 2006*; *Montgomery, Peck & Vining, 2021*), to more technical studies (*Sykes, 1993*; *Lane, 2002*) or articles outlining practical applications (*Draper & Smith, 1998*; *Rawlings, Pantula & Dickey, 2001*; *Chatterjee & Hadi, 2015*), including handbooks (*Chatterjee & Simonoff, 2013*) or works covering specific key subtopics (*Seber & Lee, 2012*). However, the reference landscape is far wider: the aforementioned considerations stimulated a steady flow of studies investigating more philosophically oriented arguments (*Allen, 2004*; *Berk, 2004*), or deeper analysis of implications related to learning (*Bartlett et al., 2020*). Given the aforementioned overall considerations, it comes as no surprise that, similarly to what happened for binary classification, a plethora of performance metrics have been defined and are currently in use for evaluating the quality of a regression model (*Shcherbakov et al., 2013*; *Hyndman & Koehler, 2006*; *Botchkarev, 2018b*, *Botchkarev, 2018a*, *Botchkarev, 2019*). The parallel with classification goes even further: in the scientific community, a shared consensus on a preferential metric is indeed far from being reached, concurring to making comparison of methods and results a daunting task.

The present study provides a contribute towards the detection of critical factors in the choice of a suitable performance metric in regression analysis, through a comparative overview of two measures of current widespread use, namely the coefficient of determination and the symmetric mean absolute percentage error.

Indeed, despite the lack of a concerted standard, a set of well established and preferred metrics does exist and we believe that, as *primus inter pares*, the coefficient of determination $R$-squared deserves a major role. The coefficient of determination is also known as R-*squared* or $R^2$ in the scientific literature. For consistency, we will use all these three names interchangeably in this study.

Introduced by *Wright (1921)* and generally indicated by $R^2$, its original formulation quantifies how much the dependent variable is determined by the independent variables, in terms of proportion of variance. Again, given the age and diffusion of $R^2$, a wealth of studies about it has populated the scientific literature of the last century, from general references detailing definition and characteristics (*Di Bucchianico, 2008*; *Barrett, 2000*; *Brown, 2009*; *Barrett, 1974*), to more refined interpretative works (*Saunders, Russell & Crabb, 2012*; *Hahn, 1973*; *Nagelkerke, 1991*; *Ozer, 1985*; *Cornell & Berger, 1987*; *Quinino, Reis & Bessegato, 2013*); efforts have been dedicated to the treatment of particular cases (*Allen, 1997*; *Blomquist, 1980*; *Piepho, 2019*; *Srivastava, Srivastava & Ullah, 1995*; *Dougherty, Kim & Chen, 2000*; *Cox & Wermuth, 1992*; *Zhang, 2017*; *Nakagawa, Johnson &*

*Schielzeth, 2017*; *Menard, 2000*) and to the proposal of *ad-hoc* variations (*Young, 2000*; *Renaud & Victoria-Feser, 2010*; *Lee et al., 2012*).

Parallel to the model explanation expressed as the variance, another widely adopted family of measures evaluate the quality of fit in terms of distance of the regressor to the actual training points. The two basic members of such family are the mean average error (MAE) (*Sammut & Webb, 2010a*) and the mean squared error (MSE) (*Sammut & Webb, 2010b*), whose difference lies in the evaluating metric, respectively linear $L_1$ or quadratic $L_2$. Once more, the available references are numerous, related to both theoretical (*David & Sukhatme, 1974*; *Rao, 1980*; *So et al., 2013*) and applicative aspects (*Allen, 1971*; *Farebrother, 1976*; *Gilroy, Hirsch & Cohn, 1990*; *Imbens, Newey & Ridder, 2005*; *Köksoy, 2006*; *Sarbishei & Radecka, 2011*). As a natural derivation, the square root of mean square error (RMSE) has been widely adopted (*Nevitt & Hancock, 2000*; *Hancock & Freeman, 2001*; *Applegate et al., 2003*; *Kelley & Lai, 2011*) to standardize the units of measures of MSE . The different type of regularization imposed by the intrinsic metrics reflects on the relative effectiveness of the measure according to the data structure. In particular, as a rule of thumb, MSE is more sensitive to outliers than MAE; in addition to this general note, several further considerations helping researchers in choosing the more suitable metric for evaluating a regression model given the available data and the target task can be drawn (*Chai & Draxler, 2014*; *Willmott & Matsuura, 2005*; *Wang & Lu, 2018*). Within the same family of measures, the mean absolute percentage error (MAPE) (*De Myttenaere et al., 2016*) focuses on the percentage error, being thus the elective metric when relative variations have a higher impact on the regression task rather than the absolute values. However, MAPE is heavily biased towards low forecasts, making it unsuitable for evaluating tasks where large errors are expected (*Armstrong & Collopy, 1992*; *Ren & Glasure, 2009*; *De Myttenaere et al., 2015*). Last but not least, the symmetric mean absolute percentage error (SMAPE) (*Armstrong, 1985*; *Flores, 1986*; *Makridakis, 1993*) is a recent metric originally proposed to solve some of the issues related to MAPE. Despite the yet not reached agreement on its optimal mathematical expression (*Makridakis & Hibon, 2000*; *Hyndman & Koehler, 2006*; *Hyndman, 2014*; *Chen, Twycross & Garibaldi, 2017*), SMAPE is progressively gaining momentum in the machine learning community due to its interesting properties (*Maiseli, 2019*; *Kreinovich, Nguyen & Ouncharoen, 2014*; *Goodwin & Lawton, 1999*),

An interesting discrimination among the aforementioned metrics can be formulated in terms of their output range. The coefficient of determination is upper bounded by the value 1, attained for perfect fit; while $R^2$ is not lower bounded, the value 0 corresponds to (small perturbations of) the trivial fit provided by the horizontal line $y = K$ for $K$ the mean of the target value of all the training points. Since all negative values for $R^2$ indicate a worse fit than the average line, nothing is lost by considering the unit interval as the meaningful range for $R^2$. As a consequence, the coefficient of determination is invariant for linear transformations of the independent variables' distribution, and an output value close to one yields a good prediction regardless of the scale on which such variables are measured (*Reeves, 2021*). Similarly, also SMAPE values are bounded, with the lower bound 0% implying a perfect fit, and the upper bound 200% reached when all the predictions

and the actual target values are of opposite sign. Conversely, MAE, MSE, RMSE and MAPE output spans the whole positive branch of the real line, with lower limit zero implying a perfect fit, and values progressively and infinitely growing for worse performing models. By definition, these values are heavily dependent on the describing variables' ranges, making them incomparable both mutually and within the same metric: a given output value for a metric has no interpretable relation with a similar value for a different measure, and even the same value for the same metric can reflect deeply different model performance for two distinct tasks (*Reeves, 2021*). Such property cannot be changed even if projecting the output into a bounded range through a suitable transformation (for example, arctangent or rational function). Given these interpretability issues, here we concentrate our comparative analysis on $R^2$ and SMAPE, both providing a high score only if the majority of the ground truth training points has been correctly predicted by the regressor. Showing the behaviour of these two metrics in several use cases and in two biomedical scenarios on two datasets made of electronic health records, the coefficient of determination is demonstrated to be superior to SMAPE in terms of effectiveness and informativeness, thus being the recommended general performance measure to be used in evaluating regression analyses.

The manuscript organization proceeds as follows. After this Introduction, in the Methods section we introduce the cited metrics, with their mathematical definition and their main properties, and we provide a more detailed description of $R^2$ and SMAPE and their extreme values ("Methods"). In the following section Results and Discussion, we present the experimental part ("Results and Discussion"). First, we describe five synthetic use cases, then we introduce and detail the Lichtinghagen dataset and the Palechor dataset of electronic health records, together with the different applied regression models and the corresponding results. We complete that section with a discussion of the implication of all the obtained outcomes. In the Conclusions section, we draw some final considerations and future developments ("Conclusions").

## METHODS

In this section, we first introduce the mathematical background of the analyzed rates ("Mathematical Background"), then report some relevant information about the coefficient of determination and SMAPE ("*R*-squared and SMAPE").

### Mathematical background

In the following formulas, $X_i$ is the predicted $i^{th}$ value, and the $Y_i$ element is the actual $i^{th}$ value. The regression method predicts the $X_i$ element for the corresponding $Y_i$ element of the ground truth dataset. Define two constants: the mean of the true values

$$\bar{Y} = \frac{1}{m} \sum_{i=1}^{m} Y_i \tag{1}$$

and the mean total sum of squares

$$\text{MST} = \frac{1}{m} \sum_{i=1}^{m} (Y_i - \bar{Y})^2 \qquad (2)$$

### Coefficient of determination ($R^2$ or $R$-squared)

$$R^2 = 1 - \frac{\sum_{i=1}^{m} (X_i - Y_i)^2}{\sum_{i=1}^{m} (\bar{Y} - Y_i)^2} \qquad (3)$$

(worst value = $-\infty$; best value = $+1$)

The coefficient of determination (*Wright, 1921*) can be interpreted as the proportion of the variance in the dependent variable that is predictable from the independent variables.

### Mean square error (MSE)

$$\text{MSE} = \frac{1}{m} \sum_{i=1}^{m} (X_i - Y_i)^2 \qquad (4)$$

(best value = 0; worst value = $+\infty$)

MSE can be used if there are outliers that need to be detected. In fact, MSE is great for attributing larger weights to such points, thanks to the $L_2$ norm: clearly, if the model eventually outputs a single very bad prediction, the squaring part of the function magnifies the error.

Since $R^2 = 1 - \frac{\text{MSE}}{\text{MST}}$ and since MST is fixed for the data at hand, $R^2$ is monotonically related to MSE (a negative monotonic relationship), which implies that an ordering of regression models based on $R^2$ will be identical (although in reverse order) to an ordering of models based on MSE or RMSE.

### Root mean square error (RMSE)

$$\text{RMSE} = \sqrt{\frac{1}{m} \sum_{i=1}^{m} (X_i - Y_i)^2} \qquad (5)$$

(best value = 0; worst value = $+\infty$)

The two quantities MSE and RMSE are monotonically related (through the square root). An ordering of regression models based on MSE will be identical to an ordering of models based on RMSE.

### Mean absolute error (MAE)

$$\text{MAE} = \frac{1}{m} \sum_{i=1}^{m} |X_i - Y_i| \qquad (6)$$

(best value = 0; worst value = $+\infty$)

MAE can be used if outliers represent corrupted parts of the data. In fact, MAE is not penalizing too much the training outliers (the $L_1$ norm somehow smooths out all the

errors of possible outliers), thus providing a generic and bounded performance measure for the model. On the other hand, if the test set also has many outliers, the model performance will be mediocre.

**Mean absolute percentage error (MAPE)**

$$\text{MAPE} = \frac{1}{m} \sum_{i=1}^{m} \left| \frac{Y_i - X_i}{Y_i} \right| \tag{7}$$

(best value = 0; worst value = $+\infty$)

MAPE is another performance metric for regression models, having a very intuitive interpretation in terms of relative error: due to its definition, its use is recommended in tasks where it is more important being sensitive to relative variations than to absolute variations (*De Myttenaere et al., 2016*). However, its has a number of drawbacks, too, the most critical ones being the restriction of its use to strictly positive data by definition and being biased towards low forecasts, which makes it unsuitable for predictive models where large errors are expected (*Armstrong & Collopy, 1992*).

**Symmetric mean absolute percentage error (SMAPE)**

$$\text{SMAPE} = \frac{100\%}{m} \sum_{i=1}^{m} \frac{|X_i - Y_i|}{(|X_i| + |Y_i|)/2} \tag{8}$$

(best value = 0; worst value = 2)

Initially defined by *Armstrong (1985)*, and then refined in its current version by *Flores (1986)* and *Makridakis (1993)*, SMAPE was proposed to amend the drawbacks of the MAPE metric. However, there is little consensus on a definitive formula for SMAPE, and different authors keep using slightly different versions (*Hyndman, 2014*). The original SMAPE formula defines the maximum value as 200%, which is computationally equivalent to 2. In this manuscript, we are going to use the first value for formal passages, and the second value for numeric calculations.

**Informativeness**

The rates RMSE, MAE, MSE and SMAPE have value 0 if the linear regression model fits the data perfectly, and positive value if the fit is less than perfect. Furthermore, the coefficient of determination has value 1 if the linear regression model fits the data perfectly (that means if MSE = 0), value 0 if MSE = MST, and negative value if the mean squared error, MSE, is greater than mean total sum of squares, MST.

Even without digging into the mathematical properties of the aforementioned statistical rates, it is clear that it is difficult to interpret sole values of MSE, RMSE, MAE and MAPE, since they have $+\infty$ as upper bound. An MSE = 0.7, for example, does not say much about the overall quality of a regression model : the value could mean both an excellent regression model and a poor regression model . We cannot know it unless the maximum MSE value for the regression task is provided or unless the distribution of all the ground truth values is known. The same concept is valid for the other rates having $+\infty$ as upper bound, such as RMSE, MAE and MAPE.

The only two regression scores that have strict real values are the non-negative $R$-squared and SMAPE. $R$-squared can have negative values, which mean that the regression performed poorly. $R$-squared can have value 0 when the regression model explains none of the variability of the response data around its mean (*Minitab Blog Editor, 2013*).

The positive values of the coefficient of determination range in the [0, 1] interval, with 1 meaning perfect prediction. On the other side, the values of SMAPE range in the [0, 2], with 0 meaning perfect prediction and 2 meaning worst prediction possible.

This is the main advantage of the coefficient of determination and SMAPE over RMSE, MSE, MAE, and MAPE: values like $R^2 = 0.8$ and SMAPE = 0.1, for example, clearly indicate a very good regression model performance, regardless of the ranges of the ground truth values and their distributions. A value of RMSE, MSE, MAE, or MAPE equal to 0.7, instead, fails to inform us about the quality of the regression performed.

This property of $R$-squared and SMAPE can be useful in particular when one needs to compare the predictive performance of a regression on two different datasets having different value scales. For example, suppose we have a mental health study describing a predictive model where the outcome is a depression scale ranging from 0 to 100, and another study using a different depression scale, ranging from 0 to 10 (*Reeves, 2021*). Using $R$-squared or SMAPE we could compare the predictive performance of the two studies without making additional transformations. The same comparison would be impossible with RMSE, MSE, MAE, or MAPE.

Given the better robustness of $R$-squared and SMAPE over the other four rates, we focus the rest of this article on the comparison between these two statistics.

## *R*-squared and SMAPE

### *R*-squared

The coefficient of determination can take values in the range $(-\infty, 1]$ according to the mutual relation between the ground truth and the prediction model. Hereafter we report a brief overview of the principal cases.

$R^2 \geq 0$: With linear regression with no constraints, $R^2$ is non-negative and corresponds to the square of the multiple correlation coefficient.

$R^2 = 0$: The fitted line (or hyperplane) is horizontal. With two numerical variables this is the case if the variables are independent, that is, are uncorrelated. Since $R^2 = 1 - \frac{\text{MSE}}{\text{MST}}$, the relation $R^2 = 0$ is equivalent to MSE = MST, or, equivalently, to:

$$\sum_{i=1}^{m} (Y_i - \bar{Y})^2 = \sum_{i=1}^{m} (Y_i - X_i)^2 \tag{9}$$

Now, Eq. 9 has the obvious solution $X_i = \bar{Y}$ for $1 \leq i \leq m$, but, being just one quadratic equation with $m$ unknowns $X_i$, it has infinite solutions, where $X_i = \bar{Y} \pm \varepsilon_i$ for a small $\varepsilon_i$, as shown in the following example:

- $\{Y_i \ 1 \leq i \leq 10\}$ = {90.317571, 40.336481, 5.619065,44.529437, 71.192687, 32.036909, 6.977097, 66.425010, 95.971166, 5.756337}

- $\bar{Y} = 45.91618$

- $\{X_i \ 1 \le i \le 10\} = \{45.02545, 43.75556, 41.18064, 42.09511, 44.85773, 44.09390, 41.58419, 43.25487, 44.27568, 49.75250\}$

- MSE = MST = 1051.511

- $R^2 \approx 10^{-8}$ .

$R^2 < 0$: This case is only possible with linear regression when either the intercept or the slope are constrained so that the "best-fit" line (given the constraint) fits worse than a horizontal line, for instance if the regression line (hyperplane) does not follow the data (*CrossValidated, 2011b*). With nonlinear regression, the $R$-squared can be negative whenever the best-fit model (given the chosen equation, and its constraints, if any) fits the data worse than a horizontal line. Finally, negative $R^2$ might also occur when omitting a constant from the equation, that is, forcing the regression line to go through the point (0,0).

A final note. The behavior of the coefficient of determination is rather independent from the linearity of the regression fitting model: $R^2$ can be very low even for completely linear model, and vice versa, a high $R^2$ can occur even when the model is noticeably non-linear. In particular, a good global $R^2$ can be split in several local models with low $R^2$ (*CrossValidated, 2011a*).

**SMAPE**

By definition, SMAPE values range between 0% and 200%, where the following holds in the two extreme cases:

SMAPE = 0: The best case occurs when SMAPE vanishes, that is when

$$\frac{100\%}{m} \sum_{i=1}^{m} \frac{|X_i - Y_i|}{(|X_i| + |Y_i|)/2} = 0$$

equivalent to

$$\sum_{i=1}^{m} \frac{|X_i - Y_i|}{(|X_i| + |Y_i|)/2} = 0$$

and, since the $m$ components are all positive, equivalent to

$$\frac{|X_i - Y_i|}{|X_i| + |Y_i|} = 0 \ \forall \ 1 \le i \le m$$

and thus $X_i = Y_i$, that is, perfect regression.

SMAPE = 2: The worst case SMAPE = 200% occurs instead when

$$\frac{100\%}{m} \sum_{i=1}^{m} \frac{|X_i - Y_i|}{(|X_i| + |Y_i|)/2} = 2$$

equivalent to

$$\sum_{i=1}^{m} \frac{|X_i - Y_i|}{|X_i| + |Y_i|} = m$$

By the triangle inequality $|a + c| \le |a| + |c|$ computed for $b = -c$, we have that $|a - b| \le |a| + |b|$ and thus $\frac{\{|a-b|}{|a|+|b|\le1}$. This yields that SMAPE = 2 if $\frac{|X_i-Y_i|}{|X_i|+|Y_i|} = 1$ for all $i = 1,\ldots,m$. Thus we reduced to compute when $\xi(a, b) = \frac{|a-b|}{|a|+|b|} = 1$: we analyse now all possible cases, also considering the symmetry of the relation with respect to $a$ and $b$, $\xi(a,b) = \xi(b,a)$.

If $a = 0$, $\xi(0, b) = \frac{|0-b|}{|0|+|b|} = 1$ if $b = 0$.

Now suppose that $a,b > 0$: $\xi(a,a) = 0$, so we can suppose $a > b$, thus $a = b + \varepsilon$, with $a,b,\varepsilon > 0$. Then $\xi(a, b) = \xi(b + \varepsilon, \varepsilon) = \frac{\varepsilon}{2b+\varepsilon} < 1$. Same happens when $a,b < 0$: thus, if ground truth points and the prediction points have the same sign, SMAPE will never reach its maximum value.

Finally, suppose that $a$ and $b$ have opposite sign, for instance $a > 0$ and $b < 0$. Then $b = -c$, for $c > 0$ and thus $\xi(a, b) = \xi(a, -c) = \frac{|a+c|}{|a|+|c|} = \frac{a+c}{a+c} = 1$.

Summarising, SMAPE reaches its worst value 200% if

- $X_i = 0$ and $Y_i = 0$ for all $i = 1,\ldots,m$

- $X_i = 0$ and $Y_i = 0$ for all $i = 1,\ldots,m$

- $X_i \cdot Y_i < 0$ for all $i = 1,\ldots,m$, that is, ground truth and prediction always have opposite sign, regardless of their values.

For instance, if the ground truth points are $(1, -2, 3, -4, 5, -6, 7, -8, 9, -10)$, any prediction vector with all opposite signs (for example, $(-307.18, 636.16, -469.99, 671.53, -180.55, 838.23, -979.18, 455.16, -8.32, 366.80)$) will result in a SMAPE metric reaching 200%.

Explained the extreme cases of $R$-squared and SMAPE, in the next section we illustrate some significant, informative use cases where these two rates generate discordant outcomes.

## RESULTS AND DISCUSSION

In this section, we first report some particular use cases where we compare the results of $R$-squared and SMAPE ("Use Cases"), and then we describe a real biomedical scenario where the analyzed regression rates generate different rankings for the methods involved ("Medical Scenarios").

As mentioned earlier, we exclude MAE, MSE, RMSE and MAPE from the selection of the best performing regression rate. These statistics range in the $[0, +\infty)$ interval, with 0 meaning perfect regression, and their values alone therefore fail to communicate the quality of the regression performance, both on good cases and in bad cases. We know for example that a negative coefficient of determination and a SMAPE equal to 1.9 clearly correspond to a regression which performed poorly, but we do not have a specific value for MAE, MSE, RMSE and MAPE that indicates this outcome. Moreover, as mentioned earlier, each value of MAE, MSE, RMSE and MAPE communicates the quality of the regression only relatively to other regression performances, and not in an absolute manner, like $R$-squared and SMAPE do. For these reasons, we focus on the coefficient of determination and SMAPE for the rest of our study.

## Use cases

We list hereafter a number of example use cases where the coefficient of determination and SMAPE produce divergent outcomes, showing that $R^2$ is more robust and reliable than SMAPE, especially on bad poor quality regressions. To simplify comparison between the two measures, define the complementary normalized SMAPE as:

$$\text{cnSMAPE} = 1 - \frac{\text{SMAPE}}{200\%} \tag{10}$$

(worst value = 0; best value = 1)

### UC1 use case

Consider the ground truth set $REAL = \{r_i = (i, i) \in \mathbb{R}^2, \ i \in \mathbb{N}, 1 \leq i \leq 100\}$ collecting 100 points with positive integer coordinates on the straight line $y = x$. Define then the set $PRED_j = \{p_i\}$ as

$$p_i = \begin{cases} r_i & \text{if } i \not\equiv 1 (\text{mod} 5) \\ r_{5k+1} & \text{for } k \geq j \\ 0 & \text{for } i = 5k+1 \,, 0 \leq k < j \end{cases} \tag{11}$$

so that REAL and $PRED_j$ coincides apart from the first $j$ points 1, 6, 11,… congruent to 1 modulo 5 that are set to 0. Then, for each $5 \leq j \leq 20$, compute $R^2$ and cnSMAPE (Table 1).

Both measures decrease with the increasing number of non-matching points $p_{5k+1} = 0$, but cnSMAPE decreases linearly, while $R^2$ goes down much faster, better showing the growing unreliability of the predicted regression. At the end of the process, $j = 20$ points out of 100 are wrong, but still cnSMAPE is as high as 0.80, while $R^2$ is 0.236, correctly declaring $PRED_{20}$ a very weak prediction set.

### UC2 use case

In a second example, consider again the same REAL dataset and define the three predicting sets

$$PRED_{start} = \{p_i^s : 1 \leq i \leq 100\}$$

$$p_i^s = \begin{cases} r_i & \text{for } i \geq 10 \\ 0 & \text{for } i < 10 \end{cases}$$

$$PRED_{middle} = \{p_i^m : 1 \leq i \leq 100\}$$

$$p_i^m = \begin{cases} r_i & \text{for } i \leq 50 \text{ and } i \geq 61 \\ 0 & \text{for } 51 \leq i \leq 60 \end{cases}$$

$$PRED_{end} = \{p_i^e : 1 \leq i \leq 100\}$$

$$p_i^e = \begin{cases} r_i & \text{for } i \leq 90 \\ 0 & \text{for } i \geq 91 \end{cases}$$

In all the three cases *start*, *middle*, *end* the predicting set coincides with REAL up to 10 points that are set to zero, at the beginning, in the middle and at the end of the prediction, respectively. Interestingly, cnSMAPE is 0.9 in all the three cases, showing that SMAPE is sensible only to the number of non-matching points, and not to the magnitude of the predicting error. $R^2$ instead correctly decreases when the zeroed sequence of points is

**Table 1 UC1 use case.**

| j | $R^2$ | cnSMAPE |
|---|---|---|
| 5 | 0.9897 | 0.9500 |
| 6 | 0.9816 | 0.9400 |
| 7 | 0.9701 | 0.9300 |
| 8 | 0.9545 | 0.9200 |
| 9 | 0.9344 | 0.9100 |
| 10 | 0.9090 | 0.9000 |
| 11 | 0.8778 | 0.8900 |
| 12 | 0.8401 | 0.8800 |
| 13 | 0.7955 | 0.8700 |
| 14 | 0.7432 | 0.8600 |
| 15 | 0.6827 | 0.8500 |
| 16 | 0.6134 | 0.8400 |
| 17 | 0.5346 | 0.8300 |
| 18 | 0.4459 | 0.8200 |
| 19 | 0.3465 | 0.8100 |
| 20 | 0.2359 | 0.8000 |

**Note:**
Values generated through Eq. (11). $R^2$, coefficient of determination (Eq. (3)). cnSMAPE, complementary normalized SMAPE (Eq. (10)).

further away in the prediction and thus farthest away from the actual values: $R^2$ is 0.995 for $PRED_{start}$, 0.6293 for $PRED_{middle}$ and $-0.0955$ for $PRED_{end}$.

**UC3 use case**

Consider now the as the ground truth the line $y = x$, and sample the set $T$ including twenty positive integer points $T = \{t_i = (x_i, y^T_i) = (i,i)\ 1 \leq i \leq 20\}$ on the line. Define $REAL = \{r_i = (x_i, y^R_i) = (i, i + N(i))\ 1 \leq i \leq 20\}$ as the same points of $T$ with a small amount of noise $N(i)$ on the $y$ axes, so that $r_i$ are close but not lying on the $y = x$ straight line. Consider now two predicting regression models:

- The set $PRED_c = T$ representing the correct model;
- The set $PRED_w$ representing the (wrong) model with points defined as $p^w_i = f(x_i)$, for $f$ the 10-th degree polynomial exactly passing through the points $r_i$ for $1 \leq i \leq 10$.

Clearly, $p^w_i$ coincides with $r_i$ for $1 \leq i \leq 10$, but $||p^w_i - r_i||$ becomes very large for $i \geq 11$. On the other hand $t_i \neq r_i$ for all $i$'s, but $||t_i - r_i||$ is always very small. Compute now the two measures $R^2$ and cnSMAPE on the first $N$ points $i = 1, \ldots, N$ for $2 \leq N \leq 20$ of the two different regression models $c$ and $w$ with respect to the ground truth set REAL (Table 2).

For the correct regression model, both measures are correctly showing good results. For the wrong model, both measures are optimal for the first 10 points, where the prediction exactly matches the actual values; after that, $R^2$ rapidly decreases supporting the inconsistency of the model, while cnSMAPE is not affected that much, arriving for $N = 20$ to a value 1/2 as a minimum, even if the model is clearly very bad in prediction.

**Table 2 UC3 use case.**

| N | Correct model R$^2$ | cnSMAPE | Wrong model R$^2$ | cnSMAPE |
|---|---|---|---|---|
| 2 | −16.1555357 | 0.3419595 | 1 | 1 |
| 3 | −0.1752271 | 0.5177952 | 1 | 1 |
| 4 | 0.7189524 | 0.6118408 | 1 | 1 |
| 5 | 0.7968514 | 0.6640983 | 1 | 1 |
| 6 | 0.8439391 | 0.7162407 | 1 | 1 |
| 7 | 0.8711581 | 0.7537107 | 1 | 1 |
| 8 | 0.8777521 | 0.7772273 | 1 | 1 |
| 9 | 0.9069923 | 0.7962306 | 1 | 1 |
| 10 | 0.9196087 | 0.8101526 | 1 | 1 |
| 11 | 0.9226216 | 0.8230926 | $-2.149735 \times 10^2$ | 0.9090909 |
| 12 | 0.9379797 | 0.8362582 | $-1.309188 \times 10^4$ | 0.8333333 |
| 13 | 0.9439415 | 0.8447007 | $-2.493881 \times 10^5$ | 0.7692308 |
| 14 | 0.9475888 | 0.8518829 | $-2.752456 \times 10^6$ | 0.7142857 |
| 15 | 0.9551004 | 0.8613108 | $-2.276742 \times 10^7$ | 0.6666667 |
| 16 | 0.9600758 | 0.8679611 | $-1.391877 \times 10^8$ | 0.6250000 |
| 17 | 0.9622725 | 0.8740207 | $-7.457966 \times 10^8$ | 0.5882353 |
| 18 | 0.9607997 | 0.8784127 | $-3.425546 \times 10^9$ | 0.5555556 |
| 19 | 0.9659541 | 0.8837482 | $-1.275171 \times 10^{10}$ | 0.5263158 |
| 20 | 0.9635534 | 0.8870441 | $-4.583919 \times 10^{10}$ | 0.5000000 |

**Note:**
We define N, correct model, and wrong model in the UC3 Use case paragraph. R$^2$, coefficient of determination (Eq. (3)). cnSMAPE, complementary normalized SMAPE (Eq. (10)).

## UC4 use case

Consider the following example: the seven actual values are $(1, 1, 1, 1, 1, 2, 3)$, and the predicted values are $(1, 1, 1, 1, 1, 1, 1)$. From the predicted values, it is clear that the regression method worked very poorly: it predicted 1 for all the seven values.

If we compute the coefficient of determination and SMAPE here, we obtain $R$-squared = −0.346 and SMAPE = 0.238. The coefficient of determination illustrates that something is completely off, by having a negative value. On the contrary, SMAPE has a very good score, that corresponds to 88.1% correctness in the cnSMAPE scale.

In this use case, if a inexperienced practitioner decided to check only the value of SMAPE to evaluate her/his regression, she/he would be misled and would wrongly believe that the regression went 88.1% correct. If, instead, the practitioner decided to verify the value of $R$-squared, she/he would be alerted about the poor quality of the regression. As we saw earlier, the regression method predicted 1 for all the seven ground truth elements, so it clearly performed poorly.

## UC5 use case

Let us consider now a vector of 5 integer elements having values $(1, 2, 3, 4, 5)$, and a regression prediction made by the variables $(a, b, c, d, e)$. Each of these variables can assume all the integer values between 1 and 5, included. We compute the coefficient of determination and cnSMAPE for each of the predictions with respect to the actual values.

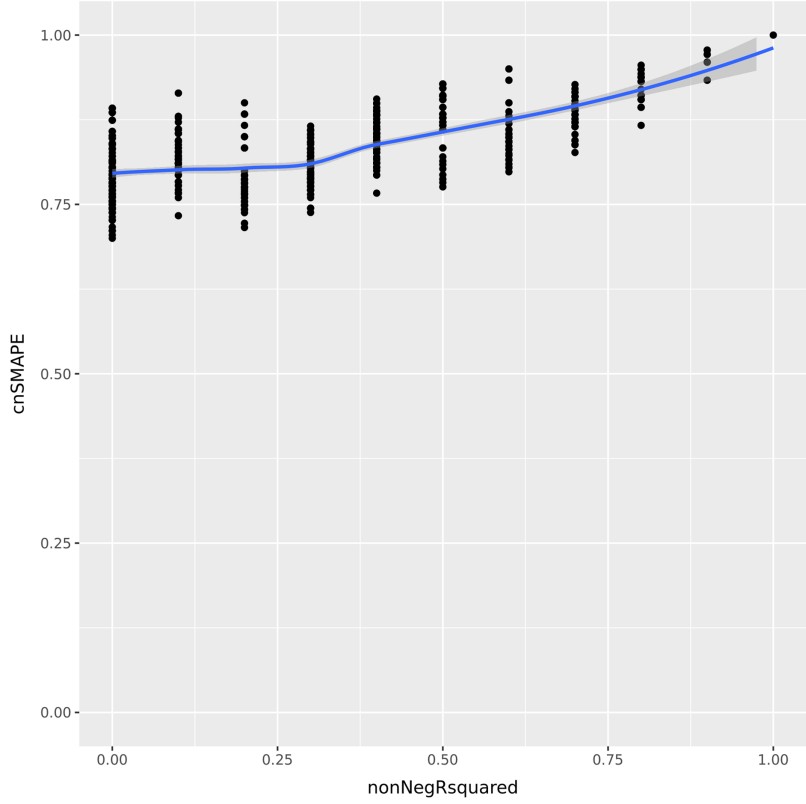

**Figure 1 UC5 Use case: R-squared versus cnSMAPE.** Representation plot of the values of cnSMAPE (Eq. (10)) on the y axis and non-negative R-squared (Eq. (3)) on the x axis, obtained in the UC5 Use case. Blue line: regression line generated with the loess smooth method.

To compare the values of the coefficient of determination and cnSMAPE in the same range, we consider only the cases when $R$-squared is greater or equal to zero, and we call it non-negative $R$-squared. We reported the results in Fig. 1.

As clearly observable in the plot Fig. 1, there are a number of points where cnSMAPE has a high value (between 0.6 and 1) but $R$-squared had value 0: in these cases, the coefficient of determination and cnSMAPE give discordant outcomes. One of these cases, for example, is the regression where the predicted values have values $(1, 2, 3, 5, 2)$, $R^2 = 0$, and cnSMAPE = 0.89.

In this example, cnSMAPE has a very high value, meaning that the prediction is 89% correct, while $R^2$ is equal to zero. The regression correctly predicts the first three points $(1, 2, 3)$, but fails to classify the forth element (4 is wrongly predicted as 5), and the fifth element (5 is mistakenly labeled as 2). The coefficient of determination assigns a bad outcome to this regression because it fails to correctly classify the only members of the 4 and 5 classes. Diversely, SMAPE assigns a good outcome to this prediction because the variance between the actual values and the predicted values is low, in proportion to the overall mean of the values.

Faced with this situation, we consider the outcome of the coefficient of determination more reliable and trustworthy: similarly to the Matthews correlation coefficient (MCC)

(*Matthews, 1975*) in binary classification (*Chicco & Jurman, 2020*; *Chicco, Tötsch & Jurman, 2021*; *Tötsch & Hoffmann, 2021*; *Chicco, Starovoitov & Jurman, 2021*; *Chicco, Warrens & Jurman, 2021*), *R*-squared generates a high score only if the regression is able to correctly classify most of the elements of each class. In this example, the regression fails to classify all the elements of the 4 class and of the 5 class, so we believe a good metric would communicate this key-message.

## Medical scenarios

To further investigate the behavior of *R*-squared, MAE, MAPE, MSE, RMSE and SMAPE, we employed these rates to a regression analysis applied to two real biomedical applications.

### Hepatitis dataset

We trained and applied several machine learning regression methods on the Lichtinghagen dataset (*Lichtinghagen et al., 2013*; *Hoffmann et al., 2018*), which consists of electronic health records of 615 individuals including healthy controls and patients diagnosed with cirrhosis, fibrosis, and hepatitis. This dataset has 13 features, including a numerical variable stating the diagnosis of the patient, and is publicly available in the *University of California Irvine Machine Learning Repository (2020)*. There are 540 healthy controls (87.8%) and 75 patients diagnosed with hepatitis C (12.2%). Among the 75 patients diagnosed with hepatitis C, there are: 24 with only hepatitis C (3.9%); 21 with hepatitis C and liver fibrosis (3.41%); and 30 with hepatitis C, liver fibrosis, and cirrhosis (4.88%).

### Obesity dataset

To further verify the effect of the regression rates, we applied the data mining methods to another medical dataset made of electronic health records of young patients with obesity (*Palechor & De-La-Hoz-Manotas, 2019*, *De-La-Hoz-Correa et al., 2019*). This dataset is publicly available in the *University of California Irvine Machine Learning Repository (2019)* too, and contains data of 2,111 individuals, with 17 variables for each of them. A variable called *NObeyesdad* indicates the obesity level of each subject, and can be employed as a regression target. In this dataset, there are 272 children with insufficient weight (12.88%), 287 children with normal weight (13.6%), 351 children with obesity type I (16.63%), 297 children with obesity type II (14.07%), 324 children with obesity type III (15.35%), 290 children with overweight level I (13.74%), and 290 children with overweight level II (13.74%). The original curators synthetically generated part of this dataset (*Palechor & De-La-Hoz-Manotas, 2019*, *De-La-Hoz-Correa et al., 2019*).

### Methods

For the regression analysis, we employed the same machine learning methods two of us authors used in a previous analysis (*Chicco & Jurman, 2021*): Linear Regression (*Montgomery, Peck & Vining, 2021*), Decision Trees (*Rokach & Maimon, 2005*), and Random Forests (*Breiman, 2001*), all implemented and executed in the R programming language (*Ihaka & Gentleman, 1996*). For each method execution, we first shuffled the patients data, and then we randomly selected 80% of the data elements for the training set and used the remaining

**Table 3 Regression results on the prediction of hepatitis, cirrhosis, and fibrosis from electronic health records, and corresponding rankings based on rates.**

|  | $R^2$ | MAE | MSE | SMAPE | RMSE | MAPE |
|---|---|---|---|---|---|---|
| Random forests (RF) | **0.756** | 0.149 | 0.133 | 1.808 | 0.361 | 0.092 |
| Linear regression (LR) | **0.535** | 0.283 | 0.260 | 1.840 | 0.498 | 0.197 |
| Decision tree (DT) | **0.423** | 0.157 | 0.311 | 0.073 | 0.546 | 0.080 |
| **Rankings:** |  |  |  |  |  |  |
| 1st | RF | RF | RF | DT | RF | DT |
| 2nd | LR | DT | LR | RF | LR | RF |
| 3rd | DT | LR | DT | LR | DT | LR |

Note:
We performed the analysis on the Lichtinghagen dataset (*Lichtinghagen et al., 2013*; *Hoffmann et al., 2018*) with the methods employed by *Chicco & Jurman (2021)*. We report here the average values achieved by each method in 100 executions with 80% randomly chosen data elements used for the training set and the remaining 20% used for the test set. $R^2$: worst value $-\infty$ and best value $+1$. SMAPE: worst value 2 and best value 0. MAE, MAPE, MSE and RMSE: worst value $+\infty$ and best value 0. We reported the complete regression results including the standard deviations in Table S1. $R^2$ formula: Eq. (3). MAE formula: Eq. (6). MAPE formula: Eq. (7). MSE formula: Eq. (4). RMSE formula: Eq. (5). SMAPE formula: Eq. (8). We highlighted the values of the coefficient of determination in bold.

20% for the test set. We trained each method model on the training set, applied the trained model to the test set, and saved the regression results measured through $R$-squared, MAE, MAPE, MSE, RMSE, and SMAPE. For the hepatitis dataset , we imputed the missing data with the Predictive Mean Matching (PMM) approach through the Multiple Imputation by Chained Equations (MICE) method (*Buuren & Groothuis-Oudshoorn, 2010*). We ran 100 executions and reported the results means and the rankings based on the different rates in Table 3 (hepatitis dataset) and in Table 4 (obesity dataset).

**Hepatitis dataset results: different rate, different ranking**

We measured the results obtained by these regression models on the Lichtinghagen hepatitis dataset with all the rates analyzed in our study: $R^2$, MAE, MAPE, RMSE, MSE and SMAPE (lower part of Table 3).

These rates generate 3 different rankings. $R^2$, MSE and RMSE share the same ranking (Random Forests, Linear Regression and Decision Tree). SMAPE and MAPE share the same ranking (Decision Tree, Random Forests and Linear Regression). MAE has its own ranking (Random Forests, Decision Tree and Linear Regression).

It is also interesting to notice that these six rates select different methods as top performing method. $R^2$, MAE, MSE and RMSE indicate Random Forests as top performing regression model, while SMAPE and MAPE select Decision Tree for the first position in their rankings. The position of Linear Regression changes, too: on the second rank for $R^2$, MSE and RMSE, while on the last rank for MAE, SMAPE and MAPE.

By comparing all these different standings, a machine learning practitioner could wonder what is the most suitable rate to choose, to understand how the regression experiments actually went and which method outperformed the others. As explained earlier, we suggest the readers to focus on the ranking generated by the coefficient of determination, because it is the only metric that considers the distribution of all the ground truth values, and generates a high score only if the regression correctly predict most of the values of each ground truth category. Additionally, the fact that the ranking indicated

**Table 4 Regression results on the prediction of obesity level from electronic health records, and corresponding rankings based on rates.**

| Method | R² | MAE | MSE | SMAPE | RMSE | MAPE |
|---|---|---|---|---|---|---|
| Random forests (RF) | **0.865** | 0.412 | 0.512 | 0.087 | 0.714 | 0.094 |
| Decision tree (DT) | **0.426** | 1.214 | 2.170 | 0.326 | 1.471 | 0.286 |
| Linear regression (LR) | **0.254** | 1.417 | 2.828 | 0.296 | 1.681 | 0.325 |
| **Rankings:** | | | | | | |
| 1st | RF | RF | RF | RF | RF | RF |
| 2nd | DT | DT | DT | LR | DT | DT |
| 3rd | LR | LR | LR | DT | LR | LR |

Note:
Mean values and standard deviations out of 100 executions with 80% randomly chosen data elements used for the training set and the remaining 20% used for the test set. We performed the analysis on the Palechor dataset (*Palechor & De-La-Hoz-Manotas, 2019*; *De-La-Hoz-Correa et al., 2019*) with the methods Linear Regression, Decision Tree and Random Forests. We report here the average values achieved by each method in 100 executions with 80% randomly chosen data elements used for the training set and the remaining 20% used for the test set. $R^2$ : worst value $-\infty$ and best value $+1$. SMAPE: worst value 2 and best value 0. MAE, MAPE, MSE, and RMSE: worst value $+\infty$ and best value 0. We reported the complete regression results including the standard deviations in Table S2. $R^2$ formula: Eq. (3). MAE formula: Eq. (6). MAPE formula: Eq. (7). MSE formula: Eq. (4). RMSE formula: Eq. (5). SMAPE formula: Eq. (8). We highlighted the values of the coefficient of determination in bold.

by *R*-squared (Random Forests, Linear Regression and Decision Tree) was the same standing generated by 3 rates out of 6 suggests that it is the most informative one (Table 3).

**Hepatitis dataset results: R² provides the most informative outcome**

Another interesting aspect of these results on the hepatitis dataset regards the comparison between coefficient of determination and SMAPE (Table 3). We do not compare the standing of *R*-squared with MAE, MSE, RMSE, and MAPE because these four rates can have infinite positive values and, as mentioned earlier, this aspect makes it impossible to detect the quality of a regression from a single score of these rates.

*R*-squared indicates a very good result for Random Forests ($R^2 = 0.756$), and good results for Linear Regression ($R^2 = 0.535$) and Decision Tree ($R^2 = 0.423$). On the contrary, SMAPE generates an excellent result for Decision Tree (SMAPE = 0.073), meaning almost perfect prediction, and poor results for Random Forests (SMAPE = 1.808) and Linear Regression (SMAPE = 1.840), very close to the upper bound (SMAPE = 2) representing the worst possible regression.

These values mean that the coefficient of determination and SMAPE generate discordant outcomes for these two methods: for *R*-squared, Random Forests made a very good regression and Decision Tree made a good one; for SMAPE, instead, Random Forests made a catastrophic regression and Decision Tree made an almost perfect one. At this point, a practitioner could wonder which algorithm between Random Forests and Decision Trees made the better regression. Checking the standings of the other rates, we clearly see that Random Forests resulted being the top model for 4 rates out of 6, while Decision Tree resulted being the worst model for 3 rates out of 6. This information confirms that the ranking of *R*-squared is more reliable than the one of SMAPE (Table 3).

**Obesity dataset results: agreement between rankings, except for SMAPE**

Differently from the rankings generated on the hepatitis dataset, the rankings produced on the obesity dataset are more concordant (Table 4). Actually, the ranking of the

coefficient of determination, MSE, RMSE, MAE and MAPE are identical: Random Forests on the first position, Decision Tree on the second position, and Linear Regression on the third and last position. All the rates' rankings indicate Random Forests as the top performing method.

The only significant difference can be found in the SMAPE standing: differently from the other rankings that all put Decision Tree as second best regressor and Linear Regression as worst regressor, the SMAPE standing indicates Linear Regression as runner-up and Decision Tree on the last position. SMAPE, in fact, swaps the positions of these two methods, compared to $R$-squared and the other rates: SMAPE says Linear Regression outperformed Decision Tree, while the other rates say that Decision Tree outperformed Linear Regression.

Since five out of six rankings confirm that Decision Tree generated better results than Linear Regression, and only one of six say vice versa, we believe that is clear that the ranking indicated by the coefficient of determination is more informative and trustworthy than the ranking generated by SMAPE.

## CONCLUSIONS

Even if regression analysis makes a big chunk of the whole machine learning and computational statistics domains, no consensus has been reached on a unified prefered rate to evaluate regression analyses yet. In this study, we compared several statistical rates commonly employed in the scientific literature for regression task evaluation, and described the advantages of $R$-squared over SMAPE, MAPE, MAE, MSE and RMSE.

Despite the fact that MAPE, MAE, MSE and RMSE are commonly used in machine learning studies , we showed that it is impossible to detect the quality of the performance of a regression method by just looking at their singular values. An MAPE of 0.7 alone, for example, fails to communicate if the regression algorithm performed mainly correctly or poorly. This flaw left room only for $R^2$ and SMAPE. The first one has negative values if the regression performed poorly, and values between 0 and 1 (included) if the regression was good. A positive value of $R$-squared can be considered similar to percentage of correctness obtained by the regression. SMAPE, instead, has the value 0 as best value for perfect regressions and has the value 2 as worst value for disastrous ones.

In our study, we showed with several use cases and examples that $R^2$ is more truthful and informative than SMAPE: $R$-squared, in fact, generates a high score only if the regression correctly predicted most of the ground truth elements for each ground truth group, considering their distribution. SMAPE, instead, focuses on the relative distance between each predicted value and its corresponding ground truth element, without considering their distribution. In the present study SMAPE turned out to perform bad in identifying bad regression models.

A limitation of $R^2$ arises in the negative space. When $R$-squared has negative values, it indicates that the model performed poorly but it is impossible to know how bad a model performed. For example, an $R$-squared equal to −0.5 alone does not say much about the quality of the model, because the lower bound is −∞. Differently from SMAPE that

has values between 0 and 2, the minus sign of the coefficient of determination would however clearly inform the practitioner about the poor performance of the regression.

Although regression analysis can be applied to an infinite number of different datasets, with infinite values, we had to limit the present to a selection of cases, for feasibility purposes. The selection of use cases presented here are to some extent limited, since one could consider infinite many other use cases that we could not analyze here. Nevertheless, we did not find any use cases in which SMAPE turned out to be more informative than $R$-squared. Based on the results of this study and our own experience, $R$-squared seems to be the most informative rate in many cases, if compared to SMAPE, MAPE, MAE, MSE and RMSE. We therefore suggest the employment of $R$-squared as the standard statistical measure to evaluate regression analyses, in any scientific area.

In the future, we plan to compare $R^2$ with other regression rates such as Huber metric $H_\delta$ (*Huber, 1992*), LogCosh loss (*Wang et al., 2020*) and Quantile $Q_\gamma$ (*Yue & Rue, 2011*). We will also study some variants of the coefficient of determination, such as the adjusted $R$-squared (*Miles, 2014*) and the coefficient of partial determination (*Zhang, 2017*). Moreover, we will consider the possibility to design a brand new metric for regression analysis evaluation, that could be even more informative than $R$-squared.

## LIST OF ABBREVIATIONS

**COVID-19** Coronavirus disease 2019
**DT** Decision Trees
**LR** Linear Regression
**MAE** Mean absolute error
**MAPE** Mean absolute percentage error
**MSE** Mean square error
$R^2$ $R$-squared, coefficient of determination
**RF** Random Forests
**RMSE** root mean square error
**SMAPE** symmetric mean absolute percentage error

## ACKNOWLEDGEMENTS

The authors thank David Reeves (the University of Manchester) for his useful advice.

### Funding

The authors received no funding for this work.

### Competing Interests

Davide Chicco is an Academic Editor for PeerJ Computer Science.

## Author Contributions

- Davide Chicco conceived and designed the experiments, performed the experiments, analyzed the data, performed the computation work, prepared figures and/or tables, authored or reviewed drafts of the paper, and approved the final draft.
- Matthijs J. Warrens analyzed the data, authored or reviewed drafts of the paper, contributed to the analysis of the mathematical properties, and approved the final draft.
- Giuseppe Jurman conceived and designed the experiments, performed the experiments, analyzed the data, performed the computation work, prepared figures and/or tables, authored or reviewed drafts of the paper, and approved the final draft.

## Data Availability

The software code is available at GitHub: https://github.com/davidechicco/R-squared_versus_other_regression_rates.

## Supplemental Information

Supplemental information for this article can be found online at http://dx.doi.org/10.7717/peerj-cs.623#supplemental-information.

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
