# Peer review of "The coefficient of determination R-squared is more informative than SMAPE, MAE, MAPE, MSE and RMSE in regression analysis evaluation"

_PeerJ Computer Science, doi:10.7717/peerj-cs.623_

## Round 0.1 · original submission · Major Revisions

We have received a mixed review report. Two of the reviewers are positive and one is negative. Please revise the paper according to all three reviewers' comments and provide point to point responses.

·

Basic reporting

No comment.

Experimental design

No comment.

Validity of the findings

No comment.

Additional comments

-The reviewer attaches a PDF document with some additional comments, suggestions and highlighted typos.
-The authors split their data into train and test sets. It would be interesting to see results with train, validation and test sets, for instance.

Reviewer 2 ·

Basic reporting

The manuscript follows the defined standards. Includes a thorough review of the literature, and contextualisation of the work within the state-of-the-art. Methods and experiments are described thoroughly. Results are clearly and transparently presented. Data is either provided or freely available online (Lichtinghagen dataset). Language is very good, with only some typos/mistakes to correct (e.g. "deeper description of for R2" on line 122, "despite of the ranges" on line 191).

Experimental design

The choice of R2 and SMAPE is well-justified based on informativeness, among all the other surveyed metrics. The experiments cover most aspects where R2 and SMAPE differ, especially the use-cases 1-5. The real medical scenario is interesting, but the manuscript would be much more complete if other real scenarios were included: other medical scenarios (e.g., COVID-19 cases prediction) or other real contexts (e.g., financial predictions).

Validity of the findings

The findings are valid. The use-cases and the medical experiment cover most differences between R2 and SMAPE and truly show the benefit of using R2. However, beyond the issue of the negative space discussed by the authors in the conclusion, no other drawbacks or possible limitations of R2 have been discussed. Perhaps additional experiments in other real scenarios or the future comparison with Huber metric Hδ, LogCosh loss, and Quantile Qγ would help unveil such downsides of R2.

Nevertheless, the lack of deeper discussion on these makes the paper appear quite one-sided (in favour of R2), and also leads to the absence of proposals to improve: what could be improved in R2, what desirable behaviours should the new R2 verify, how could we change R2 to achieve it? Since the paper is merely a comparison between R2 and SMAPE, that does not propose an improvement on either measure, at least it should include this deeper speculative discussion on what could be done to achieve an improved R2 metric.

Reviewer 3 ·

Basic reporting

Professional English: the writing of the article is mostly good professional English, with some minor errors.

Literature references: the article cited a good number of original sources of information.

Article structure: the article appears to follow common structured used for medical publications.

Self-contained: The authors did not make a clear statement about the central hypotheses of this article.

Formal results: No clear criteria was given in how to judge the different metrics.

Experimental design

The core of the work is to compare different quality metrics, however, since the authors did not establish a concrete standard, it is very hard to say which one is better.

Originality: poor.

Research question: not well defined

Rigor: anecdotal evaluation

Methods: lack of the key evaluation criteria

Validity of the findings

This is an anecdotal evaluation of five different quality metrics for regression, however, due to a lack of clear standard of comparison, it is hard to judge whether the conclusions are valid or not.

Additional comments

This article needs a clearly defined standard for comparing different quality metrics.

---

## Round 0.2 · Minor Revisions

The reviewers are generally positive about the manuscript. Please make the suggested changes and provide a point to point response.

·

Basic reporting

No comment.

Experimental design

No comment.

Validity of the findings

No comment.

Additional comments

The reviewer attaches a PDF document with some additional comments, suggestions and
highlighted typos.

Reviewer 2 ·

Basic reporting

No new comments to add.

Experimental design

No new comments to add.

Validity of the findings

No new comments to add.

Additional comments

Overall, the article is good and has been improved since the last review round.

The new scenario of children obesity prediction is exactly what I was looking for, and further illustrates the conclusions drawn by the authors regarding the superiority of R-squared.

Once again, it would be interesting to see a deeper and more insightful discussion on possible steps towards better measures, and I look forward to seeing it in future studies.

---

## Round 0.3 · accepted · Accept

All reviewers' comments have been addressed. The paper can be accepted.